METHODS

# Regularized sequence-context mutational trees capture variation in mutation rates across the human genome

Christopher J. Adams[1], Mitchell Conery[1], Benjamin J. Auerbach[1], Shane T. Jensen[2], Iain Mathieson[3], Benjamin F. Voight[3,4,5]*

1 Genomics and Computational Biology Graduate Group, Perelman School of Medicine, University of Pennsylvania, Philadelphia, Pennsylvania, United States of America, 2 Department of Statistics and Data Science, The Wharton School at the University of Pennsylvania, Philadelphia, Pennsylvania, United States of America, 3 Department of Genetics, Perelman School of Medicine, University of Pennsylvania, Philadelphia, Pennsylvania, United States of America, 4 Department of Systems Pharmacology and Translational Therapeutics, Perelman School of Medicine, University of Pennsylvania, Philadelphia, Pennsylvania, United States of America, 5 Institute for Translational Medicine and Therapeutics, Perelman School of Medicine, University of Pennsylvania, Philadelphia, Pennsylvania, United States of America

* bvoight@upenn.edu

**Data Availability Statement:** All data used for this project was obtained from the public domain, with specific links to download provided in the Supplement (S3 Text). We have implemented our

## Abstract

Germline mutation is the mechanism by which genetic variation in a population is created. Inferences derived from mutation rate models are fundamental to many population genetics methods. Previous models have demonstrated that nucleotides flanking polymorphic sites–the local sequence context–explain variation in the probability that a site is polymorphic. However, limitations to these models exist as the size of the local sequence context window expands. These include a lack of robustness to data sparsity at typical sample sizes, lack of regularization to generate parsimonious models and lack of quantified uncertainty in estimated rates to facilitate comparison between models. To address these limitations, we developed Baymer, a regularized Bayesian hierarchical tree model that captures the heterogeneous effect of sequence contexts on polymorphism probabilities. Baymer implements an adaptive Metropolis-within-Gibbs Markov Chain Monte Carlo sampling scheme to estimate the posterior distributions of sequence-context based probabilities that a site is polymorphic. We show that Baymer accurately infers polymorphism probabilities and well-calibrated posterior distributions, robustly handles data sparsity, appropriately regularizes to return parsimonious models, and scales computationally at least up to 9-mer context windows. We demonstrate application of Baymer in three ways–first, identifying differences in polymorphism probabilities between continental populations in the 1000 Genomes Phase 3 dataset, second, in a sparse data setting to examine the use of polymorphism models as a proxy for *de novo* mutation probabilities as a function of variant age, sequence context window size, and demographic history, and third, comparing model concordance between different great ape species. We find a shared context-dependent mutation rate architecture underlying our models, enabling a transfer-learning inspired strategy for modeling germline mutations. In summary, Baymer is an accurate polymorphism probability estimation

Baymer method into software that is freely available as a python package. This can be accessed on the Voight Lab GitHub repository: https://github.com/bvoightlab/Baymer. Model outputs from Baymer described in the manuscript are available at: https://doi.org/10.5281/zenodo.7843023.

**Funding:** B.F.V. acknowledges support for this work from the NIH/NIDDK (DK101478 and DK126194). The funders had no role in study design, data collection and analysis, decision to publish, or preparation of the manuscript.

**Competing interests:** The authors have declared that no competing interests exist.

algorithm that automatically adapts to data sparsity at different sequence context levels, thereby making efficient use of the available data.

## Author summary

Many biological questions rely on accurate estimates of where and how frequently mutations arise in populations. One factor that has been shown to predict the probability that a mutation occurs is the local DNA sequence surrounding a potential site for mutation. It has been shown that increasing the size of local DNA sequence immediately surrounding a site improves prediction of where, what type, and how frequently the site is mutated. However, current methods struggle to take full advantage of this trend as well as capturing how certain our estimates are, in practice. We have designed a model, implemented in software (named *Baymer*), that is able to use large windows of sequence context to accurately model mutation probabilities in a computationally efficient manner. We use Baymer to identify specific DNA sequences that have the biggest impacts on mutability and apply the model to find motifs that have potentially evolved mutability between different human populations. We also apply it to show that germline mutations observed as polymorphic sites in humans—those that have occurred in our recent evolutionary history—can model very young mutations (*de novo* mutations) as well as polymorphism observed in populations of closely related great ape species.

## Introduction

Germline mutations are the primary source of genetic variation between and within species. Quantifying where, what type, and how frequently mutations arise is therefore of fundamental importance to population genetic inference and complex trait studies. Better estimates of mutation rates improve tools designed to quantify population divergence times [1], demographic history [2], and the effects of background selection [3]. Moreover, models for the underlying *de novo* mutation rate from which burden of mutations can be statistically assessed have enabled discovery of genes [4,5] and non-coding sequences [6,7] contributing to complex disease [4,5,8,9].

Our working hypothesis is that there exists an underlying structure to the context-dependent effects that shape the mutation rate. Here, we focus on polymorphism probabilities as a proxy for the mutation rate that we hypothesize share the same context-dependent architecture subject to genetic drift, demography, selection, biased gene conversion, or additional phenomenon that operate across population history. The frequency of polymorphisms varies widely across the genome [10] and correlates with several genomic features [11–13], with new mutations caused by both exogenous and endogenous sources [14]. There is considerable evidence to suggest that local nucleotide context directly relates to the probability that a nucleotide mutates. A classic example of this is the ~14-fold higher rate of C>T transitions at methylated CpG sites, owing to spontaneous deamination of 5-methylcytosine [15–17]. Long tracts of low-complexity DNA show elevated variability in mutation rates, which is hypothesized to be the result of slippage of DNA polymerase during replication [18]. This prior work suggests that local sequence context is integral to understanding variation in polymorphism rates across the genome, and that the most predictive models will be best positioned to guide elucidation of the underlying mutational mechanisms.

Our previous work demonstrated that a sequence context window of seven nucleotides (i.e., '7-mer') provided a superior model to explain patterns of genetic variation relative to smaller windows that are commonly used (e.g., 3-mers) [19]. While an advance, this model was fundamentally limited for three reasons: *scalability*, *regularization*, and *uncertainty*. First, the size of the model–which increases by a factor of four for each nucleotide included–presents intrinsic limits both computationally and in terms of statistical power. Second, while it is straightforward to assume that every sequence context is meaningful, a more parsimonious model–informed by biological intuition–might be that only a subset of contexts contributes meaningfully to the observed variation in data. This is particularly important for inference of somatic and *de novo* mutation rates or in other data-sparse situations (e.g., across species). Finally, while our previous model provided a point estimate of the mean polymorphism probability, it did not immediately emit uncertainty resulting from multinomial variance and heterogeneity in larger sequence contexts. As sequence context sizes are expanded, there is functionally less data and thus more uncertainty in estimates, making point estimates even more unreliable. Quantifying uncertainty is also required for detecting differences in probabilities across models, for example when comparing differences in rates across populations [20–22] or at functional genomic features [23]. Ideally, a method should scale the inferred context length proportional to the amount of data and the biological signal that may be present within that data while providing uncertainty in estimated parameters and underlying probabilities.

Previous work has sought to address these challenges, though methods introduced to date do not address all limitations simultaneously. Sparsity and scalability have been tackled through a deep-learning framework [24] as well as an IUPAC-motif-based clustering approach [25] which modeled polymorphism probabilities up through 9-mers. Another method explored polymorphism probabilities up through 7-mers using DNA shape covariates to reduce the parameter space [26]. All three methods are robust and effective at measuring point estimates of polymorphism probabilities in expanded sequence contexts, however none explicitly estimates the uncertainty of these parameters. Finally, the CIPI model [27] is a Bayesian method that addresses these issues, but focuses on applications with smaller context-window motifs (5-mer) in variant settings with fewer mutation events (e.g., somatic mutations in cancer or mutations in viral genomes) and is not obviously scalable computationally to larger size context windows and sizes of contemporary population genomics data sets in humans (e.g., hundreds of millions of polymorphic sites).

Here, we develop a method that addresses all three limitations embedded in a novel model. We construct a Bayesian tree-based method that integrates sequence context window size, handles sparse data, and captures uncertainty in estimates of mutation probability via the posterior distribution. We subsequently apply our approach in multiple ways. First, we quantify differences in polymorphism probabilities between continental populations and place bounds on the effect sizes of potential undescribed context-dependent differences in the 1000 Genomes dataset [28]. Second, we explore the use of polymorphism datasets to predict *de novo* mutations. We measure the effect of population history, variant age, and sequence context size on model performance with the aim of generating a meaningful proxy to estimate the germline mutation rate. Finally, we build models of different great ape species and assess the similarity to human polymorphism models.

## Description of the method

### A tree-based sequence-context model captures variation in polymorphism probabilities

We began by developing a model to describe the hierarchical relationship of sequence context dependencies over increasing window sizes. We structured this as a rooted, tree-based graph,

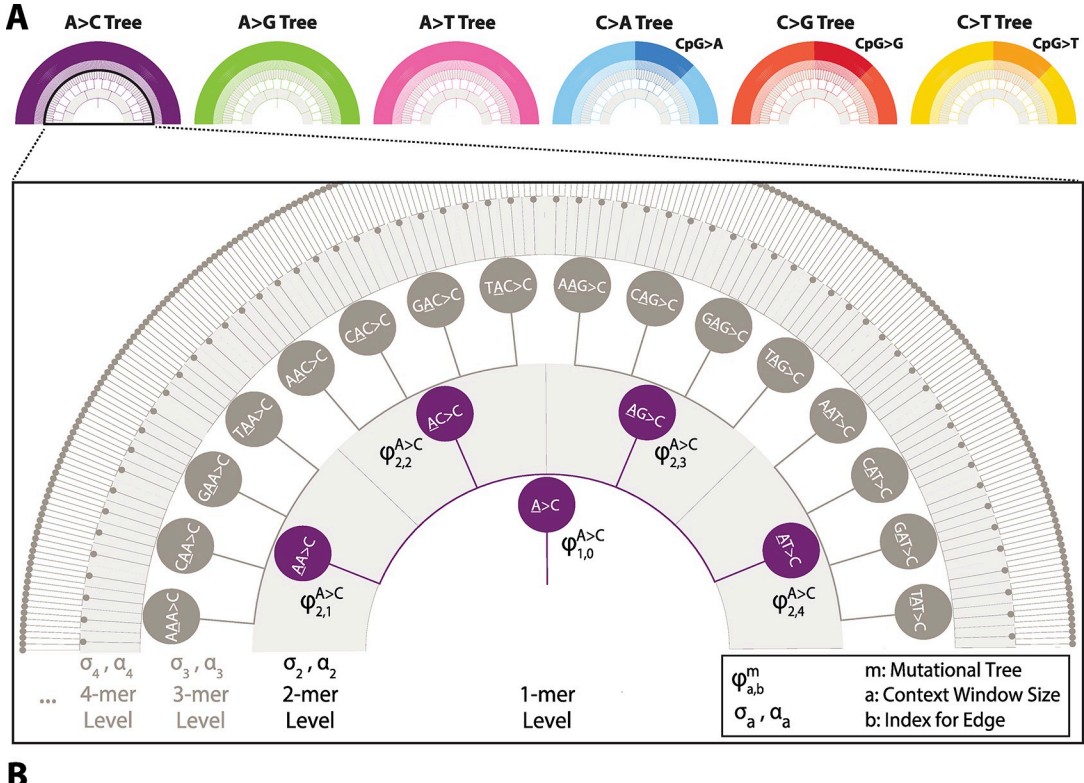

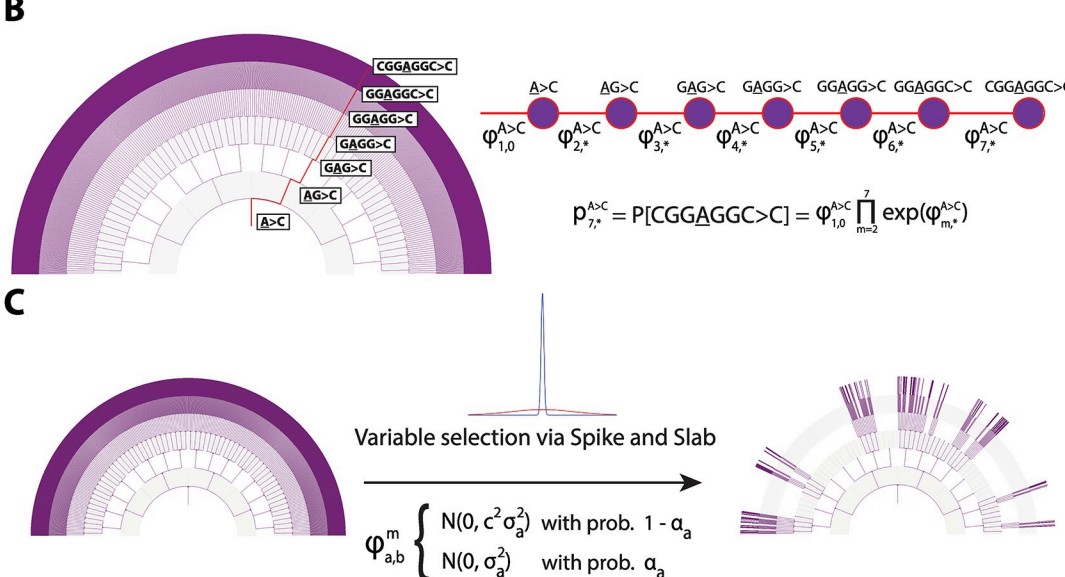

**Fig 1. Hierarchical relationship of sequence contexts and key algorithmic elements of Baymer.** (A) Each mutation type is represented by a separate sequence context tree, related by the shared 'mer' level parameters and joint multinomial likelihood distribution. Each sequence context tree has a nested structure where information is partially pooled across each shared parent. (B) Polymorphism probabilities are parameterized as the product of the series of edges that lead to the sequence context of interest. (C) Sequence context trees are regularized using a spike-and-slab prior distribution.

where each type of substitution class is represented distinctly (**Fig 1A**). Each level of the tree represents an increasing window size of sequence considered, alternating between incorporating nucleotides to the window on the 3′ end for even-sized contexts and on the 5′ end for odd-sized contexts. We fold over reverse complementary contexts to reduce the parameter count

(**S1 Text**). To ease readability, we denote each mutation with the sequence context, the nucleotide in scope underlined, and the polymorphism indicated with an arrow (e.g. TCC>T represents the polymorphism where the underlined cytosine has become a thymine). Each non-root edge represents the log-transformed, multiplicative shift in polymorphism probability captured by expanding sequence context. The root edge corresponds to an estimated base polymorphism probability for a given mutation type. For a given sequence context, each node in the tree represents the probability of observing a polymorphic site in the central nucleotide (referred to hereafter as polymorphism probability), and is the product of all edges, starting from the root that leads to the node (**Fig 1B**). As our previous work has shown for a specific level of sequence context, the distribution of observed counts for each sequence context can be modelled via independent multinomial distributions [19] facilitating likelihood calculation. The resulting multinomial probability vector corresponds to the combination of individual polymorphism probability estimates across each mutation type tree for each sequence context (**S1 Text**).

Within the model, we incorporate two features essential for downstream applications when comparing the outputs of competing models. First, we employ a Bayesian formulation which generates posterior distributions for polymorphism probabilities (**S1 Text**). This approach naturally provides uncertainty around parameter estimates which is essential for comparison of rates across different tabulated models. Second, we incorporate regularization in the parameter estimation procedure for tree edges. Previous sequence context models estimated parameters ($\phi$) for all edges of the tree, meaning that all values of were effectively non-zero. However, our previous work suggested that perhaps only a fraction of edges meaningfully contribute information [19]. Hypothesizing that only a subset of edges is informative for estimating mutation probabilities, we regularize our tree model by incorporating a *spike-and-slab* prior on the $\phi$ parameters [29]. Our approach estimates the fraction of posterior samples in the slab, implying a non-zero effect on polymorphism probabilities, and in the spike, which implies no effect. Thus, the probability of an edge being included in the slab is the equivalent of the posterior inclusion probability (PIP) for our model. We tune the model such that the slab is favored when the evidence suggests a multinomial probability shift greater than 10% for a given context level (**Fig 1C**). This value was chosen weighing the stability of model convergence with the goal of inferring the largest possible effects.

Because the posterior distribution is not analytically tractable, we implemented an adaptive Metropolis-within-Gibbs Markov Chain Monte Carlo (MCMC) sampling scheme [30] to sample from and thereby estimate the posterior distribution of this model. To further aid in convergence and enforce intermediate nodes to have identifiable mutation probabilities, we estimated parameters of the model level-by-level rather than all simultaneously, leveraging the conditional dependency structure of the hierarchical tree. Under this set-up, the unseen higher-order layers are assigned $\phi_{a,b} = 0$ edges until their level has been sampled. We embedded this model and sampling scheme into software (named Baymer) for further testing and applications.

## Verification and comparison

### Asymmetric context expansion improves parsimony and model inference

The hierarchical tree-based Baymer graphs are constructed such that the difference in length of flanking nucleotides on either side of the focal nucleotide is zero for odd-length contexts, and one for even-length contexts. It follows that these trees can be constructed in three ways: expanding sequence contexts by including even-length contexts (termed "asymmetric models")–alternating expansions starting at either the (1) left, or 5', end, or (2) starting with the right, or 3', end–or (3) expanding by exclusively using odd-length contexts (termed

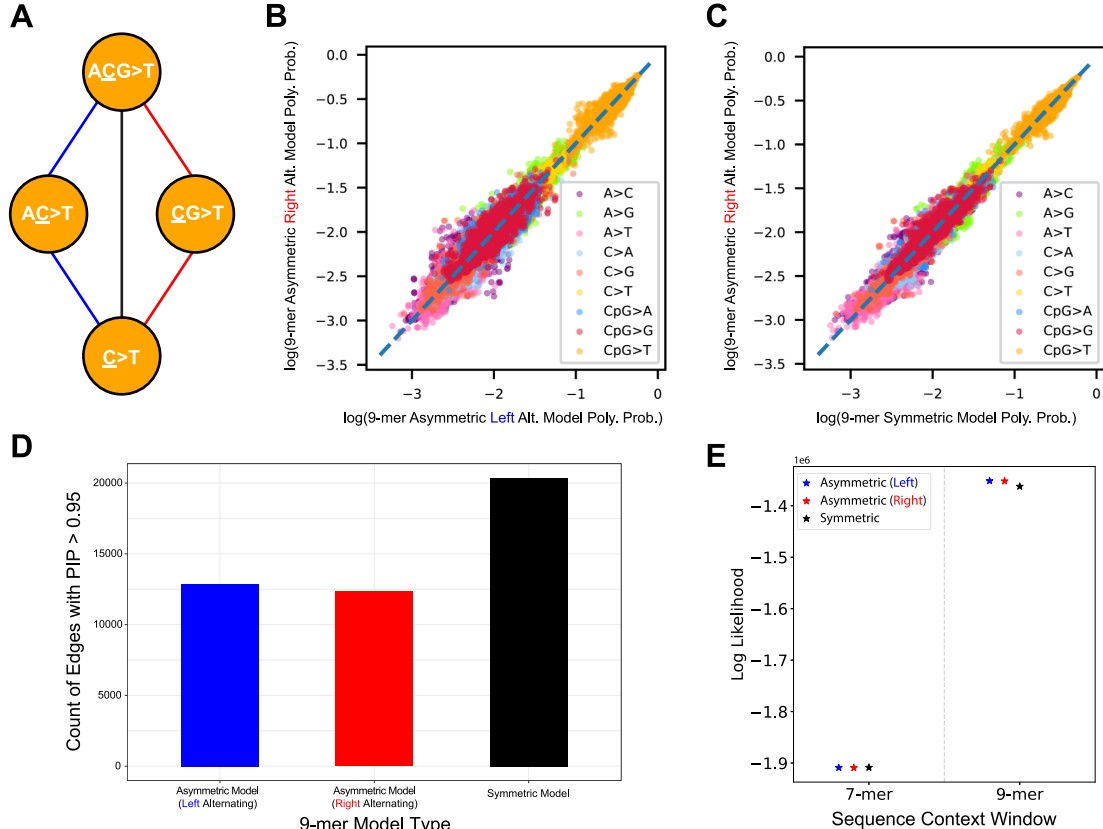

**Fig 2. Exploration of different strategies to build out the Baymer sequence context hierarchical trees using gnomAD non-Finnish European (NFE) polymorphisms with derived allele count greater than or equal to two in non-coding accessible regions.** (A) Sequence contexts can be built by starting to alternate adding nucleotides with the left, or 5', end of each odd-length context (blue path), with the right, or 3', end of each context (red path), or by only including odd-length contexts and adding a nucleotide to both sides of the growing context (black path) for each expansion. (B) Baymer mean posterior estimates of 9-mer polymorphism probabilities estimated using even base pair data with the right alternation pattern and the left alternation pattern (Spearman correlation = 0.997; p < 10$^{-100}$). (C) Baymer mean posterior estimates of 9-mer polymorphism probabilities estimated using even base pair data with the right alternation pattern and a model exclusively using odd-length sequence contexts (Spearman correlation = 0.998; p < 10$^{-100}$). (D) Absolute count of edges in each tree architecture model with a PIP > 0.95 in a 9-mer model. (E) Multinomial likelihoods for each model are calculated on odd base pair NFE test data using 7-mer and 9-mer models. Polymorphism probability estimates were linearly scaled to match the mean polymorphism probability of the holdout dataset.

"symmetric models"; **Fig 2A**). When expanding an odd-length context by two nucleotides (e.g., 1-mer to 3-mer), symmetric models require 16 edges per context as compared to 20 edges per context (16 edges + 4 intermediate edges) in the asymmetric model. Despite more total edges in the asymmetric model tree architecture, we hypothesized the intermediate edges would more efficiently capture signal and provide greater resolution to detect inflection points in the tree where the local sequence context results in mutability changes. We implemented and tested all three models on the same dataset. First, we observed that for our most challenging model (9-mers), rate estimates are very strongly correlated amongst all models (**Fig 2B and 2C**). Furthermore, we observed that despite more total edges in the tree-graph, asymmetric models include approximately 38% fewer overall edges with high confidence (**Fig 2D**), suggesting greater parsimony. Finally, asymmetric models produced models that better fit holdout data than the symmetric models (**S1 Text** and **Fig 2E**). This improvement arises specifically in situations where there is sufficient data to estimate 8-mer edges, but insufficient data to confidently estimate 9-mer rates. Given our folding scheme, we opted for the

biologically pragmatic choice of an even-length context tree architectural model that initiates alternating context expansions with the right end (3'), as this captures CpG effect(s) as early as possible in the tree without distributing the effect across more than one edge.

## Evaluation of the model demonstrates robust inference of the underlying rates with uncertainty

A key feature of Baymer is that it estimates posterior distributions for each parameter, allowing for uncertainty in the probabilities of polymorphism at each sequence context. To evaluate the coverage of the estimated posterior probabilities, we used simulations to assess how often our posterior distribution captures simulated values. Using a pre-specified polymorphism probability table, we tested how frequently polymorphism probabilities estimated by Baymer captured the true value for each sequence context (**S1 Text**). We found that across all sequence context sizes, 89%, 93%, and 97% of context simulations contained the true polymorphism probability in the 90%, 95%, and 99% credible intervals, respectively (**S1 Text and S1 Table**).

A second important feature is that regularization is embedded into the method, allowing for the creation of parsimonious models that capture most of the information with the fewest non-zero parameters. This part is critical to address cases where the amount of data is not large and limits power, or when considering larger windows of sequence context that are rare and/or uninformative. If robustly calibrated, we would expect probabilities inferred in a hold-out set to strongly correlate with those estimated during a test phase (i.e., minimal overfitting). To evaluate the robustness of the inferred rates, we partitioned the human genome reference into two sets–even and odd base pairs–and used SNPs of allele count 2 or greater observed in the gnomAD [31] non-Finnish European (NFE) collection to independently train models (**S1 Text**). We compared the concordance of probabilities for models with sequence context windows up to 4 flanking nucleotides on either side (i.e., a 9-mer model) using the maximum likelihood estimate approach [19] and Baymer (**S1 Fig**). For each comparison, in addition to the Spearman correlation, we also calculated the root mean squared perpendicular error (RMSPE) from each point to the x-y axis, as a measure of the tightness of the distribution from the true, shared value (**S1 Text**). The maximum likelihood estimates of polymorphism probabilities (**Fig 3A,** Spearman correlation $\rho = 0.915$; RMSPE = 0.117) were less correlated and considerably less tightly distributed than those for Baymer-derived models (**Fig 3B,** $\rho = 0.990$; RMSPE = 0.035). This result occurred even after omitting ~16,000 sequence contexts with zero mutations in either dataset (odd and even base pairs) from the maximum likelihood model comparison, rendering practical use of large swaths of the model useless due to substantial overfitting at the 9-mer level. If zero-mutation contexts omitted from the maximum likelihood model were included, the correlations would perform considerably worse (**S1 Text and S1D Fig,** $\rho = 0.876$; RMSPE = 0.744), as these polymorphism probabilities are exclusively determined by pseudo counts. Within the NFE dataset, Baymer inferences were also robust across allele frequency bins (**S2 Fig**).

We next sought to evaluate the transferability of inferred models between experimental collections; while internally consistent, the above procedure could simply reflect data set specific biases [32]. For this, we compared non-admixed, non-Finnish European (EUR) samples obtained from the 1000 Genomes (1KG) Project (re-sequenced by the New York Genome Center) [33] with the gnomAD NFE sample described above. As before, we split the data into even and odd base pairs but also applied a variant down-sampling procedure to match total variant count and site-frequency spectrum between both sets (**S1 Text**). By comparing variants found in the even base pair genome of gnomAD with the odd base pair genome of 1KG, this strategy ensures no variation overlapped between data sets. We observed that the probabilities

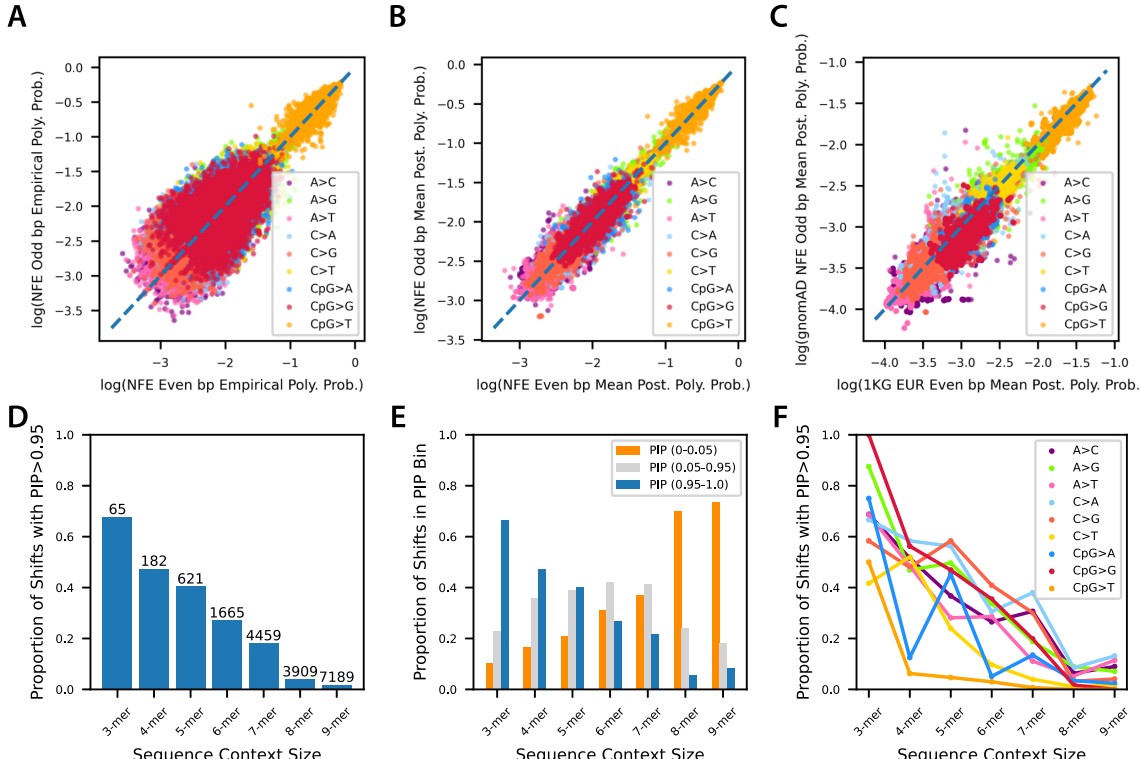

**Fig 3. Baymer model validation, transferability, and regularization in gnomAD non-Finnish European (NFE) polymorphisms with derived allele count greater than or equal to two in non-coding accessible regions.** (A) Empirical 9-mer polymorphism probabilities for context mutations with at least one occurrence in both datasets (15,910 omitted context mutations) are plotted against one another (Spearman correlation = 0.915; $p < 10^{-100}$; RMSPE = 0.12). (B) Baymer mean posterior estimates for 9-mer polymorphism estimates in even and odd base pair datasets (Spearman correlation = 0.990; $p < 10^{-100}$; RMSPE = 0.035). (C) Baymer mean posterior estimates for 9-mer polymorphism estimates in odd base pair non-Finnish European gnomAD data and even base pair NYGC 1KG phase three data, down-sampled to match total number of polymorphisms and site frequency spectrum (Spearman correlation = 0.984; $p < 10^{-100}$; RMSPE = 0.045). (D) Fraction of edges in the NFE model with a PIP > 0.95 in each sequence context window layer. Absolute count of edges above bars. (E) For high-data contexts with at least 100,000 total instances in the non-coding genome and 50 total mutations, fraction of edges at each sequence context window size across PIP bins. (F) Proportion of high-data contexts within each mutation type at each sequence context window size with PIP>0.95.

estimated from both sample sets were strongly correlated (ρ = 0.984; RMSPE = 0.045; **Fig 3C**) though were slightly weaker than the correlations from each internal comparison and fit less tightly (gnomAD ρ = 0.990; RMSPE = 0.035; **Fig 3B**; 1KG ρ = 0.986; RMSPE = 0.042; **S3 Fig**). This result demonstrates that while some additional between-sample variation may exist, Baymer infers probabilities of polymorphism that are broadly consistent with one another, supporting the notion of model transferability across different data sets.

We next aimed to quantify how well the model selects meaningful context features. We expected more proximal bases to the focal site to have a greater impact on polymorphism probabilities for two reasons, (i) due to data richness, and (ii) that proximity to the polymorphic site would suggest more direct impacts on mutability, e.g., the CpG context. Consistent with expectation, the fraction of edges with a PIP > 0.95 monotonically decreases as the sequence context size is increased (**Fig 3D**). For any given 9-mer context, we find a median of 3 edges included with high confidence in the model (**S4A Fig**). The median window of context-dependence for each 9-mer was 5 base pairs wide, although this inference is limited by the sparsity of the model (**S4B Fig**).

### Larger contexts best explain patterns of variation genome-wide

We note that over 61% of all edges with a PIP > 0.95 are found in the 8-mer and 9-mer levels of our model of polymorphism observed in the gnomAD NFE data. While fewer than 2% of 9-mer edges meaningfully impact the final estimates, they still account for the most total absolute edges (7189 total edges > 0.95 PIP) and are enriched for larger effect sizes (**S4C Fig**). This observation holds even after filters for data sparsity (**S1 Text** and **Fig 3E**). This implies a considerable impact on polymorphism probabilities in extended sequence contexts, consistent with previous work [19,23–25]. This general trend is similarly consistent across mutation types (**Fig 3F**), although with a variable degree of impact, most notably with less additional variability estimated in wider CpG>T edges (**S4D Fig**). We thus evaluated the overall improvement in likelihood by expanding window sizes up to 9-mers. Compared to lower context models (e.g., 3-mer, 5-mer, or 7-mer) on holdout data, 9-mer Baymer models substantially improved the likelihood and best fit to the data (**S1 Text** and **S2 Table**).

## Applications

### Sequence context motifs are correlated with changes in polymorphism probability

We next aimed to identify inflection points in the Baymer trees by examining the edges corresponding to the largest $\phi$s across each layer. Unsurprisingly, the CpG>T edge had the largest mean posterior $\phi$ magnitude (**S3 Table**). Consistent with our previous results [19], edges with the largest absolute mean posterior $\phi$ are largely localized at the intersection of poly-A repeat-rich sequences (lower rates of A>T substitutions) but particularly presented in 8-mer context by poly-A tract of length 4, where the mutation type extends one of the repeated patterns (e.g., CGCGAGAGA>C or CCCAAAA>C), and the CAATN motif which increases A>G mutability (2.35–2.99x increase, **S3 Table**).

Next, we attempted to discover specific motifs that are enriched in the highest or lowest 1% of 9-mer polymorphism probabilities within each mutation type (**S1 Text** and **S4 Table**). We recapitulate almost all previously reported motifs [19,23,25]. Consistent with previous reports, we identify a preponderance of repeat-rich motifs, which is perhaps due to the impact of slippage in introducing mutations [18]. We discover numerous motifs with flanks extending 4 base pairs from the focal nucleotide that showed enrichment (21 motifs with p < 0.0001; **S4 Table**), emphasizing the utility of expanded sequence context windows for modeling mutability.

### Frequency of polymorphism across populations do not differ substantially across levels of sequence context

Prior work has centered around evaluating whether mutation rates have changed over evolutionary time by evaluating differences in the proportions of sequence-context-dependent polymorphism between human populations [21,22,34–36]. To determine whether polymorphism probabilities differ across human populations, we analyzed individuals from the NYGC resequencing of 1000 Genomes Project (1KG) Phase III representing continental European, African, East Asian, and South Asian groups. We extracted variants private to these continental groups, down-sampling to match site-frequency spectra bins and overall sample sizes (**S1 Text**). We then applied Baymer to each individual dataset to model probabilities up to a 9-mer window of sequence context. We compared estimates of polymorphism probabilities in each population by assessing the degree to which the posterior distribution of each population's model parameters overlapped. The fraction overlap of each distribution is a proxy for

the probability that the underlying parameters are the same. Due to the implicit tree structure of sequence context models, polymorphism probability shifts in edges will affect all edges downstream of the context in question. Therefore, we identified contexts where both the estimated polymorphism probability and the immediate edge leading to that context were both considered very likely to be different between populations.

Specifically, we identified contexts whose posterior estimates of polymorphism probabilities and edges both overlapped less than 1% in pairwise comparisons between the four populations (S5 Table). This included all the most notable previously reported 3-mer shifts across continental groups, including the increase in TCC>T mutations found in European relative to non-European ancestry populations [20–22,34,35].

We next focused on the remainder of 3-mer and wider extended sequence contexts (Table 1). While a handful of such sequence contexts have been implicated [34], these results

**Table 1. Baymer modeled 1KG private continental context mutations with extreme polymorphism probability differences.**

| Population Comparison | Context Mutation | ln(Poly. Prob. Fraction) | Poly. Prob. Fraction Overlap | Shift Difference | Shift Fraction Overlap | Population Specificity |
|---|---|---|---|---|---|---|
| European vs. African | TCC>T[abc] | 0.291 | 0 | -0.174 | 1.4E-157 | European |
| | TCT>T[abc] | 0.136 | 1.6E-18 | -0.116 | 8.5E-16 | European |
| | GCAATTA>G | 0.569 | 4.7E-03 | -0.668 | 2.4E-03 | - |
| | TATATATC>G | -0.660 | 7.2E-03 | 0.730 | 5.6E-03 | African |
| European vs. South Asian | TCC>T[abc] | 0.112 | 1.2E-09 | -0.059 | 2.7E-03 | European |
| | TCT>T[abc] | 0.063 | 5.0E-03 | -0.066 | 2.9E-03 | European |
| | CTATA>T | -0.587 | 2.9E-03 | 0.493 | 7.3E-03 | South Asian |
| | ATCTTC>G | -0.606 | 7.6E-03 | 0.668 | 5.4E-03 | - |
| European vs. East Asian | CCC>T[abc] | 0.081 | 1.4E-03 | 0.075 | 6.6E-04 | - |
| | TCC>T[abc] | 0.312 | 0 | -0.156 | 2.4E-97 | European |
| | GCT>T | -0.064 | 5.7E-03 | 0.095 | 6.1E-05 | - |
| | TCT>T[abc] | 0.133 | 3.0E-19 | -0.102 | 9.6E-06 | European |
| | GCAACCA>G | 1.056 | 5.3E-03 | -1.104 | 5.0E-03 | - |
| | ATACCTC>A | -1.029 | 4.2E-03 | 0.830 | 5.0E-03 | East Asian |
| African vs. South Asian | TCC>T[abc] | -0.179 | 1.7E-118 | 0.115 | 3.4E-12 | - |
| | CTATA>T | -0.507 | 6.1E-03 | 0.482 | 7.4E-03 | South Asian |
| | CCCCCAG>G | -0.818 | 2.6E-03 | 0.767 | 2.7E-03 | - |
| | TATATATC>G | 0.668 | 3.3E-03 | -0.738 | 2.2E-03 | African |
| African vs. East Asian | GCT>T | -0.063 | 9.1E-03 | 0.074 | 2.2E-03 | - |
| | CTCGCG>T | 1.240 | 2.8E-03 | -1.243 | 3.6E-03 | - |
| | TAAAATA>T | -1.160 | 3.9E-03 | 1.135 | 4.8E-03 | - |
| | ATACCTC>A | -1.061 | 4.6E-03 | 0.829 | 5.7E-03 | East Asian |
| | TATATATC>G | 0.712 | 3.9E-04 | -0.748 | 1.3E-04 | African |
| East Asian vs. South Asian | TCC>T[abc] | -0.200 | 2.4E-155 | 0.097 | 5.4E-05 | - |
| | CTATA>T | -0.519 | 5.3E-03 | 0.479 | 7.8E-03 | South Asian |
| | CTCGCG>T | -1.244 | 2.0E-03 | 1.247 | 2.7E-03 | - |
| | ATACCTC>A | 0.906 | 8.5E-03 | -0.819 | 9.1E-03 | East Asia |
| | CCCCCAG>G | -0.819 | 3.8E-03 | 0.764 | 4.4E-03 | - |

[a] previously reported in Mathieson and Reich [20]

[b] previously reported in Harris and Pritchard [22]

[c] previously reported in Aikens et al [34]

are confounded by batch effects in the original 1KG sequencing data [37]. Since the data we use for our analysis is derived from the New York Genome Center resequencing project [33], we do not expect the same confounder. In our results, we observed the presence of nucleotide repeats, e.g., TA / CG dinucleotides; poly-C / poly-A in several of the divergent contexts, which could be explained by polymerase slippage [18].

While the population-specific polymorphism probabilities estimated and polymorphism counts are identical between each pairwise comparison and thus correlated, we still note that 15/28 pairwise differences are specific to a single continental group. Of these, only the two canonical European context mutation differences (TCC>T and TCT>T) are in 3-mer contexts, otherwise all are found in 5-mer and greater window sizes. In South Asian samples, we find that the mean CTATA>T polymorphism probabilities are approximately 1.6 times higher than the remaining populations and in Africans TATATATC>G is approximately 1.9 times higher. The largest population-specific effect was discovered in East Asians where ATACCTC>A polymorphism probabilities are roughly 2.7 times higher than in European, African, or South Asian models. None of these effects have been explicitly documented before.

Taken collectively, we observed relatively few instances of edges that were quantifiably different across continental groups, and those that were observed were largely confined to relatively small windows of context where we might have anticipated well-powered tests (e.g., 3- and 5-mers). To quantify the power of our specific analytic procedure for discovery and the sample size necessary to identify true differences in polymorphism probabilities, we performed simulations where true effect differences were 'spiked-in' between two populations over a range of weak to stronger effects and across a sampling of different sequence contexts (**S1 Text**). Differences in mutability between populations for this experiment are defined as the natural log of the polymorphism probabilities ratio (NLPPR) between each simulated population. This allowed us to construct credible sets of effects that we were reasonably well powered ($>80\%$) to discover (**Table 2**). Unsurprisingly, the power scaled proportional to the number of context instances, simulated mutations in the dataset, and the size of the spiked-in differences (**S5 Fig**). Notably, extremely subtle shifts (NLPPR $<= 0.01$; 0.99–1.01-fold change) were not detectable at any sequence context size. On the opposite side of the spectrum, we found that we were reasonably powered to identify shift differences where NLPPR $> 1.0$ (fold decrease $\leq 0.37$ or fold increase $\geq 2.72$) up through 5-mers and in 6-mers with large sample sizes. For reference, the TCC>T polymorphism has an NLPPR $= 0.291$ (~1.34 fold increase)– the largest difference of any 3-mer by our calculation.

In contrast, our experiment had essentially no power to discover 9-mer polymorphism probability changes and extremely limited power for 8-mers, even for large differences. Thus, there may exist large differences at these sizes that we could not reliably capture. These results are consistent with our comparisons in the real data (**Table 1**), as only differences within the detectable range at each mer-level were implicated. These power calculations suggest that, given the experiment we performed grouping all mutations together (agnostic to allele frequency or age, see **Discussion**), if any 3-mer differences greater than the TCC>T shift exist, we would have discovered these effects for a broad range of modest to very strong effects across a range of sequence contexts window sizes. This effectively sets bounds on the differences possible for this analysis scheme in this data.

## A sequence context model that captures variability in *de novo* mutational rates

Given its formulation in handling data sparsity, we next sought to apply Baymer to develop a model that best captures rates of *de novo* mutations across the genome. We took advantage of

**Table 2. Power estimates for 1KG continental private polymorphism probabilities.**

| abs(ln(adj. poly. prob / null poly. prob.)) | # Contexts Sample Size Percentile | Fraction of Contexts with >80% Power at Each 'mer' level | | | | | | |
|---|---|---|---|---|---|---|---|---|
| | | 3-mers | 4-mers | 5-mers | 6-mers | 7-mers | 8-mers | 9-mers |
| 0.01 | 0–25% | 0 | 0 | 0 | 0 | 0 | 0 | 0 |
| | 26–50% | 0 | 0 | 0 | 0 | 0 | 0 | 0 |
| | 51–75% | 0 | 0 | 0 | 0 | 0 | 0 | 0 |
| | 76–100% | 0 | 0 | 0 | 0 | 0 | 0 | 0 |
| 0.1 | 0–25% | 0.44 | 0.11 | 0 | 0 | 0 | 0 | 0 |
| | 26–50% | 0.63 | 0.04 | 0 | 0 | 0 | 0 | 0 |
| | 51–75% | 0.73 | 0.03 | 0 | 0 | 0 | 0 | 0 |
| | 76–100% | 0.58 | 0.10 | 0 | 0 | 0 | 0 | 0 |
| 0.5 | 0–25% | 1.00 | 0.92 | 0.30 | 0.21 | 0.01 | 0 | 0 |
| | 26–50% | 1.00 | 1.00 | 0.68 | 0.15 | 0.01 | 0 | 0 |
| | 51–75% | 1.00 | 1.00 | 0.76 | 0.27 | 0.02 | 0 | 0 |
| | 76–100% | 1.00 | 1.00 | 0.87 | 0.20 | 0.03 | 0 | 0 |
| 1 | 0–25% | 1.00 | 0.99 | 0.81 | 0.34 | 0.20 | 0.02 | 0 |
| | 26–50% | 1.00 | 1.00 | 1.00 | 0.73 | 0.23 | 0.06 | 0 |
| | 51–75% | 1.00 | 1.00 | 1.00 | 0.87 | 0.24 | 0.04 | 0 |
| | 76–100% | 1.00 | 1.00 | 1.00 | 0.87 | 0.37 | 0.08 | 0 |
| 1.5 | 0–25% | 1.00 | 1.00 | 0.96 | 0.61 | 0.25 | 0.08 | 0 |
| | 26–50% | 1.00 | 1.00 | 1.00 | 0.91 | 0.39 | 0.18 | 0.02 |
| | 51–75% | 1.00 | 1.00 | 1.00 | 0.99 | 0.59 | 0.20 | 0 |
| | 76–100% | 1.00 | 1.00 | 1.00 | 0.99 | 0.69 | 0.26 | 0 |

a recent collection of 2,976 WGS Icelandic trios that identified 200,435 *de novo* events[38] and, analogous to the above, we partitioned *de novo* variants into even (for training) and odd (for testing) base pairs. We observed substantial improvement in the overall likelihood in the testing set for 5-mer size windows compared to 3-mers (3-mer vs 5-mer, ΔLL = 2,144), but only minimal improvement for increasing windows sizes further (5-mer vs 9-mer, ΔLL = 265, S6 Table). Indeed, Baymer did not select any sequence context feature beyond the 5-mer level with PIP > 0.95. This is not unexpected given our approach to regularization, as the number of events in larger sequence contexts is increasingly sparse, it is desirable to only include informative contexts to avoid overfitting.

We next used Baymer to improve upon this baseline model. Previous work has demonstrated that inference of *de novo* mutational probabilities can be captured via rare variant polymorphism data obtained from population sets as a proxy [23]. We hypothesized that a partitioned set of polymorphism data based on: (i) larger sample sizes that (ii) closely matched the ancestry of the *de novo* set and (iii) focused on rare variants as a proxy to capture the most recent mutation events would generate the most transferrable model and robust rate estimates. To build variant partitions, we used variant call set data from gnomAD, focused on either a population-matched proxy (i.e., NFE, the non-Finnish European subset) or variant calls from all samples in gnomAD regardless of ancestry (i.e., ALL). For each of these, we created three partitions focused (i) exclusively on variants with one allele count (i.e., singletons; for NFE labeled NFE-1), (ii) exclusively on variants with two allele counts (i.e., doubletons; for NFE labeled NFE-2), and (iii) variants with allele count of two or greater (for NFE labeled NFE-2+). Beyond this, we also identified a set of putatively derived substitutions in the human lineage by comparing the GRCh38 human reference genome with ancestral sequences obtained from primates[39].

We applied Baymer to each variant set independently, comparing the likelihoods of each model to explain rates of *de novo* mutation in the test set after downscaling probabilities

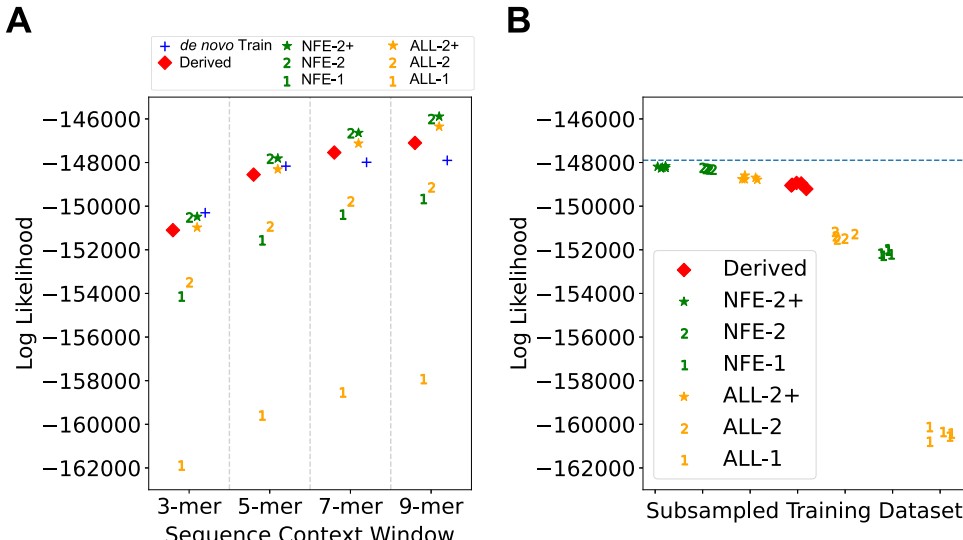

**Fig 4. Modeling de novo mutation probabilities using polymorphism datasets.** Even base pair Halldorsson et al. *de novo* training data modeled by Baymer is compared to Baymer-modelled polymorphism datasets partitioned by allele count. (A) Multinomial likelihoods for each model are calculated on odd *de novo* base pair test data at various sequence context sizes. Polymorphism probability estimates were linearly scaled to match the mean polymorphism probability of the holdout dataset. (B) Polymorphism datasets were down-sampled to match the size of the even base pair de novo data (70,364 variants) and multinomial likelihoods were calculated on odd *de novo* base pair data. Each dataset was down-sampled using 5 different random seeds. The Log Likelihood of the 9-mer *de novo* training model is indicated with the blue dotted line.

proportional to the sample size. First, we observed that for 3-mer sequence context models, the set of variants obtained from the *de novo* training set outperformed all other models despite 102 to 1,377 times fewer variants contributing to them than the polymorphism datasets (**Fig 4A and S6 Table**). In contrast, for larger windows of context (i.e., 7-mer and 9-mer), several of the polymorphism partitions explained the data better than one trained directly from *de novo* events. This result indicates that increased sample size is required to detect meaningful shifts in polymorphism probabilities in larger sequence context windows.

Despite evidence to suggest singleton datasets should best recapitulate *de novo* variation [4,23,31], we observed that models that trained exclusively on singletons and ALL-2 performed considerably worse than the rest across all windows of sequence context (**Fig 4A and S6 Table**). While our prior intuition that larger numbers of variants would have provided better rate estimates from increased power deeper in the context tree, rate models exclusively estimated with singletons suffer the most from the impact of recurrent mutations [20,40], especially at CpG sites, which include the highest polymorphism probability mutation type (CpG>T) (**S6 Table**).

Alternatively, population concordance between training and test and/or the quality of variant calls used in training the model could also impact performance. As such, we next sought to explore the effect of noise in low allele count variants. Although we only used variants that passed gnomAD quality control checks, this filter still included a large proportion of variants with a negative log-odds ratio of being a true variant (AS_VQSLOD < 0; **S6 Fig**). This pattern was also evident for other variant allele counts but were most striking in singletons and the ALL-2 variant groups. Stricter quality filters (AS_VQSLOD > 5–10) considerably improved model performance, but still did not surpass the *de novo* training model at the 3-mer level (**S6 Table**). Our NFE singleton Baymer model trained on the strictest quality filter tested

(AS_VQSLOD > 10) nearly equaled our best performing model, NFE-2+, with ~ 1/30<sup>th</sup> the number of variants, but came up just short. In summary, we observed that training from a population matched sample which excluded singletons, NFE-2+, best predicted rates of *de novo* mutations in 5-mer or larger contexts, better than models trained on *de novo* events directly.

Next, we sought to determine which sample set best modelled the *de novo* test set adjusting for the total number of variants within the partition. To control sample size differences, we downsampled each partition to match the number of variants observed in the *de novo* training set (n = 70,364) five times. After down-sampling and when considering 9-mer context models, we observed that the partitions which included NFE exclusively (noted in green, **Fig 4B**) performed on average better than using the entirety of gnomAD, "ALL" (noted in orange in **Fig 4B**), which included a more diverse panel of individuals within Europe (e.g., Finnish) but also beyond Europe (e.g., East and South Asian, African and African American). This is consistent with prior belief that, after controlling for the total sample size, variants that derive from samples where ancestries more closely match are the most informative.

## A grafted tree approach provides superior estimates of *de novo* mutational probabilities

Given the observations that *de novo* models only outperform polymorphism-based models when either small sequence contexts are used (**Fig 4A**) or the sample size is controlled (**Fig 4B**), we next sought to explore a transfer learning-inspired [41] strategy to improve upon our model performance. Transfer learning has previously been employed in a sequence context modelling setting [24]. We hypothesized that regularization means that *de novo* models have reduced performance with expanded sequence contexts due to low sample sizes. Indeed, our *de novo* model did not have the power necessary to confidently (PIP > 0.95) include any non-zero shifts in sequence contexts larger than 5-mers in the model (**Fig 5A**). The larger polymorphism datasets, however, were well-powered to detect shifts in every level of the tree (**Fig 5A**).

The nested tree structure of our polymorphism probability models provides a natural strategy where specific branches of the estimated trees can be interchanged, i.e., a "grafted" tree. We asked how similar estimates for edges in expanded sequence contexts are between our *de novo* model and the best-performing polymorphism model, NFE-2+. In edges in 2-mer and greater levels where the *de novo* training model is powered enough to detect shifts (PIP > 0.95), the mean posterior estimates of shifts are highly correlated (**Fig 5B**). This suggests a grafted tree approach is feasible, leveraging the polymorphism datasets for those edges the *de novo* model is incapable of estimating properly due to sparsity (**Fig 5C**). Therefore, we built a grafted tree model using 1- to 3-mer edges estimated in the *de novo* training data model, and 4- to 9-mer edges estimated using the NFE-2+ data model. The resulting combined model had a greater fit to the holdout *de novo* data than either the NFE-2+ model or *de novo* model alone (**Fig 5D** and **S1 Text**).

## Sequence context-dependent mutability is shared between closely related great ape species

Finally, we examined how well human polymorphism models could capture variability in polymorphism levels observed in populations of great ape. Using polymorphism data from the Great Ape Genome Project [42], we built Baymer models of *Pan troglodytes* and *Gorilla gorilla* (**S1 Text**). We note broad agreement in estimated polymorphism probabilities between humans and chimpanzees (**S7A Fig,** Spearman correlation ρ = 0.950; RMSPE = 0.089) or gorillas (**S7B Fig,** Spearman correlation ρ = 0.942; RMSPE = 0.103). These results indicate that the

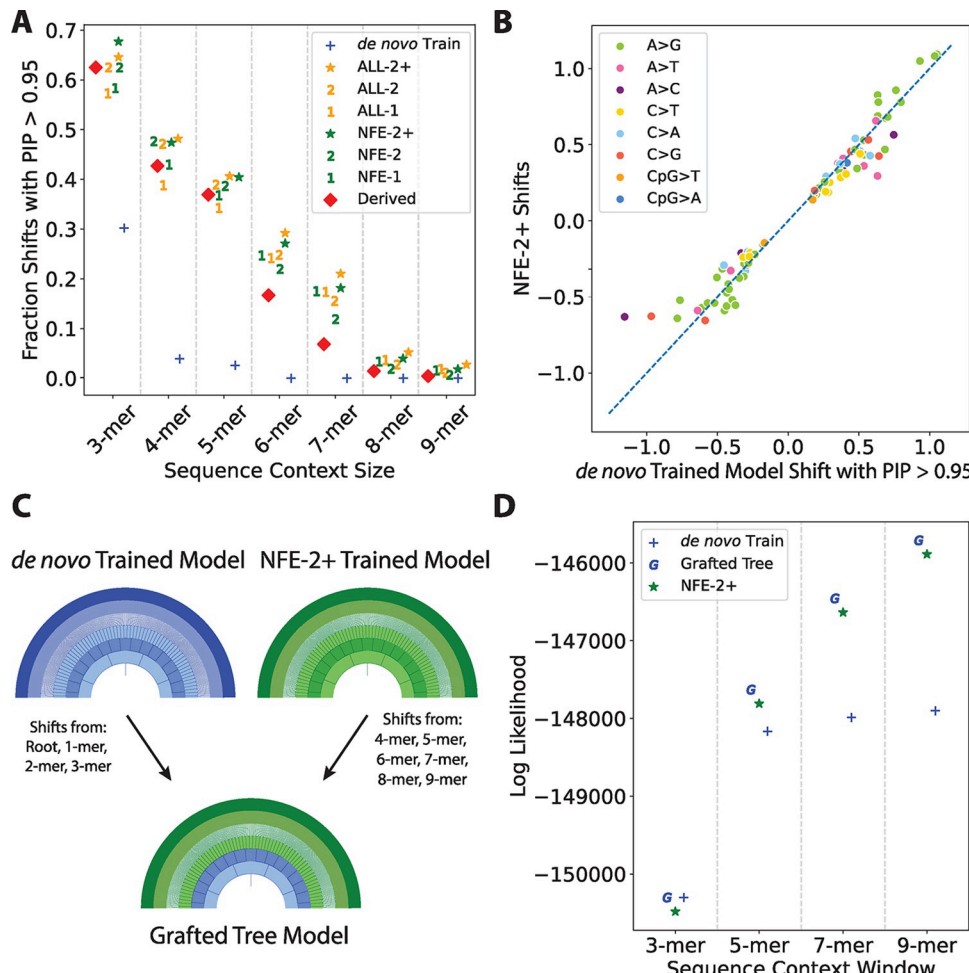

**Fig 5. Tree grafting strategy to share information between Baymer models.** (A) For each de novo proxy model, we calculated the fraction of context polymorphism probability edges with a PIP > 0.95 in 2-mer through 9 mer-levels as a proxy for the degree of regularization in each model. (B) Edges in the *de novo* training model that are included with high-confidence (PIP>0.95) are very similar in magnitude and direction to their equivalents in the best-performing proxy model, NFE-2+, in 2-mer through 9-mer levels, implying a shared polymorphism probability shift structure. (C) Proposed tree-grafting schema for modeling *de novo* mutations that leverages mer-levels where *de novo* data is plentiful (1-mer through 3-mers) and uses polymorphism data to model the remainder of each model in larger mer-levels (4-mer through 9-mers) where the *de novo* model is underpowered. (D) The grafted tree method outperforms the previously best-performing model, NFE-2+.

rates of polymorphism at higher orders of sequences contexts are similar across closely related great ape species. As we were especially interested in how human polymorphism models compared with chimpanzee and gorilla models in predicting holdout data in each respective species, we then tested models on odd base pair data in each species, training models using even base pair data for the species in focus (**S1 Text**). For both chimpanzee (**S7C Fig**) and gorilla (**S7D Fig**) tests, species-matched 9-mer models outperformed all other models. While human-derived models are outperformed at the 9-mer level, it is notable that human 9-mer models are more likely than chimpanzee 7-mer models against chimpanzee data and gorilla 5-mer models against gorilla data (**S7 Table**). Taken collectively, these results suggest the rates of polymorphism at higher orders of sequences contexts are similar across closely related great ape species, with within-species models best capturing variability in observed polymorphism levels.

## Discussion

Here, we present Baymer, a Bayesian method to model mutation rate variation that computationally scales to large windows of nucleotide sequence context (S2 Text and S8 Table), robustly manages sparse data through an efficient regularization strategy, and emits posterior probabilities that capture uncertainty in estimated probabilities. Consistent with previous studies [24–26], we show that expanded sequence context models in most current human datasets are overfit with classic empirical methods but considerably improve model performance when properly regularized. As a result, this method allows for renewed evaluation of experiments that originally were statistically limited to polymorphism probability models with small sequence context windows.

We examined differences in polymorphism probabilities between the continental populations in the 1KG project. While differences in 3-mer polymorphism probabilities have been well-documented [20–22] and expansions up to 7-mers have been tested [34], both methods rely on empirical models with frequentist measures of uncertainty. Here, we expanded the search space out to 9-mer windows and leverage the uncertainty estimated in the model to directly quantify differences in these populations. We note that many of the differences discovered contain poly-nucleotide repeats. There is some prior literature on the mechanism of slippage in polymerases during replication of such sequences [18], so differential efficiencies of these enzymes across populations could conceivably result in these patterns. However, it is also very possible that artifacts from sequencing errors with differential effects across populations could explain the differences.

Despite being well-powered to identify a large range of differences in 3-mer and smaller contexts, we identified very few contexts that differ with high probability between the populations tested. This implies that if large-scale population differences in the mutation spectrum do exist at these window context sizes, they may be comprised of numerous subtle shifts rather than a few large changes, in agreement with conclusions from prior work [22].

We also explicitly placed bounds on the magnitude of differences that could possibly exist in these data without being detected, quantifying what differences we can expect to be discovered given the way variants are grouped in this experiment. Even though the 1KG project is relatively small compared to current datasets, the number of sequence contexts available for modeling is dataset-independent and inherently limited by the sequence diversity of the human genome. Thus, while more polymorphism data could lead to the discovery of additional smaller shifts in the future, bigger datasets will not improve the power to detect larger shifts in this allele frequency agnostic setting. In fact, for very large samples, polymorphisms in some contexts can become saturated [43], reducing the information content in a similar manner as overly sparse data. Thus, both to increase power and to improve modeling resolution, it will become necessary to partition the data (e.g., by allele frequency or variant age [36], or other genomic features).

It remains a challenge to disentangle the contribution of demography [20,35,44] versus changes in the underlying mutation rate on the mutation spectrum. Here, we control for the site frequency spectrum of variants included, but the next stage of this model will need to incorporate more sophisticated demographic features. Integrating Baymer-derived trees with a joint mutation spectrum and demographic history method, such as mushi [35], is a promising future direction. While this work focuses on modeling mutability in regions minimally affected by background selection, constraint could also bias estimates. Given prior work [19] we do not expect the underlying sequence-context-mediated mutability to behave any differently in constrained regions, suggesting future Baymer-estimated codon-aware models to explicitly model expected variation in coding regions.

We also aimed to address the degree to which polymorphism datasets could be used to approximate the *de novo* mutation rate. Currently, true *de novo* mutation datasets are limited in size, which place bounds on the scope of inference for adequate sequence context modeling. We demonstrate that polymorphism datasets are accurate proxies for *de novo* mutation models and largely share the same context-dependent mutability shifts, though in contrast to reports in the literature [4,23,31], the focus exclusively on singleton variants (at least, using gnomAD calls) performed poorly relative to all other considered models. Indeed, our experiment indicates that it is preferable to use germline mutation models based on large polymorphism datasets that can estimate shifts through the 9-mer level than it is to use the largest 3-mer de novo dataset, as is commonly used in the literature [4,5,31]. Including exclusively variants from either polymorphism data or *de novo* data was also suboptimal, however, as the best possible model we built for estimating *de novo* mutation rates used *de novo* mutations in concert with polymorphism datasets. The success of this experiment implies a general context-dependent mutability architecture that underlies the human mutation spectrum. The similarity of the derived dataset, which in theory represents the oldest subset of variants tested, to the *de novo* variation further strengthens this argument. We note this dataset could in theory be biased towards European samples given the history of the Human Genome Project [45], and as such, refinements will need to be made as more diverse representations of the human genome are created. Overall, this work suggests that although there have been some well-documented small changes in context-dependent mutation rates, the general architecture remains largely conserved during modern human history.

Our experiments modeling great ape variation suggests this general architecture might be more pervasive across the tree of life. While some specific mutation spectra differences have been documented [22,46], we note broad agreement amongst closely related species as well as similar signals in extended sequence contexts. For those non-human species with WGS datasets, cohort-sizes are usually very small ($< 100$), however, Baymer is well-suited to handle these sparse data situations. Furthermore, for those species with very little data, this work opens the exciting possibility to leverage closely related species' models as priors for estimating variation in less well-characterized relatives. Further work is necessary to model species across the tree of life to determine the extent that sequence context-dependent mutability is shared and how transferable 9-mer models can be.

One limitation of the model is the treatment of multi-allelic sites. Currently, multi-allelic sites are treated as separate polymorphisms which violates assumptions of the multinomial model, where only one outcome is possible for each locus. When we excluded multi-allelic sites, we observed biases in the rates of CpG>A and CpG>G mutations, which are disproportionately filtered as a side-effect of sharing the same sequence contexts with CpG>T mutations. A more nuanced approach that models multiallelic and biallelic sites separately and then integrates jointly would deal with this issue, though multiple mutations at the same nucleotide position with the same allele change would require additional effort [40].

Finally, although we can identify regions of the tree where polymorphism probabilities diverge and thus infer critical points in the tree, this model is tailored towards polymorphism probability estimation rather than explicitly for motif discovery [27]. Our objective is to estimate polymorphism probabilities rather than finding those contexts with the largest effect sizes. Although including even-length contexts yields better-performing models, the current tree architecture only explicitly captures the effects of half of such contexts. While adding one nucleotide at a time pseudo-symmetrically for tree generation reduces the computational sampling load, it makes for more difficult interpretation of the resulting mono-nucleotide impacts. Baymer's formulation also does not model the mutability of target contexts independent of mutation type, which currently requires post-hoc analysis to identify motifs that have non-

specific mutability signatures. Future work will therefore need to integrate all possible paths through the sequence context tree and share information across contexts between mutation type trees.

In all our experiments, we focused on the entirety of the non-coding genome that is accessible to sequencing. That said, Baymer can easily be applied to any genomic features of interest for both polymorphism probability estimates and comparisons of feature-dependent sequence context mutability changes. Our approach does not currently incorporate genomic features in the model, but given genomic area bounds, polymorphism probabilities can be tailored to a biological question of interest. Addressing questions regarding the impact of genomic features on observed polymorphisms will be enhanced with well-regularized models, as smaller genomic areas or specific variant conditions can induce considerable data sparsity by reducing the number of contexts and/or polymorphisms available. Therefore, Baymer paves the way for exciting possibilities to study the effects of genomic features, variant age, and smaller subpopulations on sequence context-dependent mutation rate variation.

## Supporting information

**S1 Fig. Empirical even odd polymorphism probability scatter plots for the NFE dataset including contexts with zero mutation variants.** Baymer mean posterior estimates for (A) 3-mer models (Spearman correlation = 0.999; $p < 10^{-100}$; RMSPE = 0.0009), (B) 5-mer models (Spearman correlation = 0.999; $p < 10^{-100}$; RMSPE = 0.0063), (C) 7-mer models (Spearman correlation = 0.992; $p < 10^{-100}$; RMSPE = 0.0459), and (D) 9-mer models (Spearman correlation = 0.876; $p < 10^{-100}$; RMSPE = 0.7441) in even and odd base pair datasets. Polymorphism probabilities in the bottom two and top left quadrants correspond to those contexts where no mutations are present for the given mutation type in the respective datasets. These polymorphism probabilities are exclusively calculated using pseudocounts.
(EPS)

**S2 Fig. Comparison of Baymer mean posterior estimates for differing allele frequency (AF) bins in the NFE dataset.** (A-D) AF 0.02–0.05 compared against 0.05–0.15 AF, 0.15–0.30 AF, 0.30–0.50 AF, and 0.50–0.85 AF, respectively. (E-G) AF 0.05–0.15 AF compared against 0.15–0.30 AF, 0.30–0.50 AF, and 0.50–0.85 AF, respectively. (H-I) 0.15–0.30 AF compared against 0.30–0.50 AF and 0.50–0.85 AF, respectively. (J) 0.30–0.50 AF compared against 0.50–0.85 AF.
(EPS)

**S3 Fig. Comparison of empirical and Baymer-derived 9-mer polymorphism probabilities in NYGC-resequenced 1000 Genomes Phase 3 non-admixed non-Finnish European (EUR) polymorphisms with derived AC $\geq$ two in non-coding accessible regions.** (A) Empirical 9-mer polymorphism probabilities for context mutations with at least one occurrence in both datasets (102,875 omitted context mutations) are plotted against one another (Spearman correlation = 0.862; RMSPE = 0.175). (B) Baymer mean posterior estimates for 9-mer polymorphism estimates in even and odd base pair datasets (Spearman correlation = 0.986; RMSPE = 0.042).
(EPS)

**S4 Fig. Overview of the characteristics of edge mutability change dynamics in Baymer models of the NFE dataset.** (A) Histogram of the number of edges per 9-mer that were inferred to confidently change polymorphism probabilities (PIP > 0.95). (B) Histogram of the maximum edge size per each 9-mer that was inferred to confidently change polymorphism probabilities (PIP > 0.95). (C) Estimated distributions of phi for each mer size level. (D) The

distribution of the fractional differences of each 9-mer mean posterior polymorphism probability with their respective nested 3-mer mean posterior polymorphism probability estimates, partitioned by mutation type.
(EPS)

**S5 Fig. Fraction overlap of simulated datasets trained by Baymer at varying sequence contexts and log changes to the null polymorphism probability.**
(EPS)

**S6 Fig. Variant Quality Scores reported in gnomAD by allele count.** Distribution of gnomAD AS_VQSLOD quality scores in non-Finnish European samples ("NFE"; A-C) and in all populations ("ALL"; D-F), separated into singletons (A,D), doubletons (B,E), and variants with allele count greater than or equal to 3 (C,F).
(EPS)

**S7 Fig. Comparison of *Homo sapiens* Baymer model (NFE-2+ model) estimates with *Pan troglodytes* and *Gorilla gorilla* great ape species.** (A) Mean polymorphism estimates of *Homo sapiens* model plotted against mean polymorphism estimates of *Pan troglodytes* model (Spearman correlation = 0.957; RMSPE = 0.088). (B) Mean polymorphism estimates of *Homo sapiens* model plotted against mean polymorphism estimates of *Gorilla gorilla* model (Spearman correlation = 0.950; RMSPE = 0.097). (C) Multinomial likelihoods for each model are calculated on odd base pair *Pan troglodytes* test data at various sequence context sizes. *Pan troglodytes* model is trained on even base pair data only. (D) Multinomial likelihoods for each model are calculated on odd base pair gorilla gorilla test data at various sequence context sizes. *Gorilla gorilla* model is trained on even base pair data only. Polymorphism probability estimates were linearly scaled to match the mean polymorphism probability of the holdout dataset.
(EPS)

**S1 Table. Calibration of credible sets across mer levels by measuring number of simulations capturing the truth value.**
(XLSX)

**S2 Table. Likelihood of even base pair NFE Baymer models on 9-mer odd base pair holdout NFE data.**
(XLSX)

**S3 Table. Top 5 largest mean posterior phi estimates for each context size.**
(XLSX)

**S4 Table. Motifs tested for enrichment in the top or bottom 1% of polymorphism probabilities for each mutation type.**
(XLSX)

**S5 Table. Baymer modeled 1KG private continental context mutations with extreme polymorphism probability differences.**
(XLSX)

**S6 Table. Sample sizes and test likelihoods for each *de novo* comparison dataset.**
(XLSX)

**S7 Table. Sample sizes and test likelihoods for each great ape comparison test.**
(XLSX)

**S8 Table. Run-times for models with increasing sequence context windows.**
(XLSX)

**S1 Text. Supplementary Methods.**
(DOCX)

**S2 Text. Computational Considerations.**
(DOCX)

**S3 Text. Data Availability.**
(DOCX)

## Acknowledgments

We would like to thank Dr. Ziyue Gao for her very helpful feedback on the manuscript and in the development process.

## Author Contributions

**Conceptualization:** Christopher J. Adams, Shane T. Jensen, Iain Mathieson, Benjamin F. Voight.

**Data curation:** Christopher J. Adams, Mitchell Conery.

**Formal analysis:** Christopher J. Adams, Benjamin J. Auerbach, Benjamin F. Voight.

**Funding acquisition:** Benjamin F. Voight.

**Investigation:** Christopher J. Adams, Benjamin F. Voight.

**Methodology:** Christopher J. Adams, Benjamin F. Voight.

**Project administration:** Benjamin F. Voight.

**Resources:** Benjamin F. Voight.

**Software:** Christopher J. Adams.

**Supervision:** Benjamin F. Voight.

**Validation:** Christopher J. Adams, Benjamin F. Voight.

**Visualization:** Christopher J. Adams, Mitchell Conery, Benjamin F. Voight.

**Writing – original draft:** Christopher J. Adams, Benjamin F. Voight.

**Writing – review & editing:** Christopher J. Adams, Mitchell Conery, Benjamin J. Auerbach, Shane T. Jensen, Iain Mathieson, Benjamin F. Voight.

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
