## [Decision Letter · Decision Letter 0]

7 Feb 2023

Dear Dr Voight,

Thank you very much for submitting your Research Article entitled 'Regularized sequence-context mutational trees capture variation in mutation rates across the human genome' to PLOS Genetics.

The manuscript was fully evaluated at the editorial level and by independent peer reviewers. The reviewers appreciated the attention to an important problem, but raised some substantial concerns about the current manuscript and the associated software. Based on the reviews, we will not be able to accept this version of the manuscript, but we would be willing to review much-revised version of the manuscript with working software. We cannot, of course, promise publication at that time.

If you decide to revise the manuscript for further consideration at PLOS Genetics, please aim to resubmit within the next 60 days, unless it will take extra time to address the concerns of the reviewers, in which case we would appreciate an expected resubmission date by email to plosgenetics@plos.org.

We are sorry that we cannot be more positive about your manuscript at this stage. Please do not hesitate to contact us if you have any concerns or questions.

Yours sincerely,

Adam Siepel

Guest Editor

PLOS Genetics

David Balding

Section Editor

PLOS Genetics

Reviewer's **Comments to the Authors:**

Reviewer #1: In this manuscript, Adams et al describe a method to model the likelihood of mutations occuring in the human germline given the (up to) 9-mer nucleotide context of the mutated locus. The method incorporates a form of regularization (via the spike-and-slab prior) helping it deal with sparse/noisy data due to low mutation counts and thus lessens overfitting, and further provides estimates of uncertainty. The hierarchical nature of the model allows for a certain form of transfer-learning-like strategy. The modelling method is well described, and reasonably rigorously tested on out-of-sample data and it is convincing that it can model the extended, up to nonanucleotide, context effects on mutation rates in the human germline well.

With respect to novel biological insight gained, this study is in my opinion somewhat limited. All the analysis is on human germline mutation/polymorphism. We see that there is signal in the 9-mer context in human population variation, extended from the 7-mer reported in a previous paper from the same lab (DOI: 10.1038/ng.3511); this is arguably a bit of an incremental advance. There are reports about differences („shifts“) in mutable oligonucleotides between human populations, with again largely unclear novelty compared to another previous paper from the same lab (doi: 10.1093/molbev/msz023). In all fairness, here, because of the new method, we do have the ability to check the variation in the 9mer contexts (wider than before) however significant differences appear very rare (or, with very slight effect, so they 'slip under the radar').

There are some technical tests on what sort of polymorphisms (singletons, doubletons, rare variants, species divergence) represent the best proxy for de novo mutations, which is interesting as an approach to boost the accuracy of the methodology, but it is not clear what new biology is/can be revealed using the improvements. In this study, the method is applied in a specialized way, only to the population variants and de novo variants, without demonstrating more general application beyond population genomics. This would be shown by illustrative analyses of feasibility to apply to other data types, e.g. somatic mutations, species divergence (except as a baseline, which was implemented), or population variation in other species). Or, as touched upon in the discussion, application to different genomic regions („areas“) would be of much interest, since it can add very interesting context to germline variation analysis (e.g. DNA replication time associates with mutagenesis quite strongly, and is often considered in genome-wide mutagenesis analyses). Thus the paper also misses, in my opinion, showcasing of the method on more diverse set of applications, to demonstrate broad interest.

Some specific suggestions follow:

1. Lines 182-184, describing the concordance test: "If zero-mutation contexts omitted from the maximum likelihood model were included, the correlations would perform considerably worse (Methods, Supplementary Fig. 1D, ρ = 0.876; RMSPE = 0.744)". Could the authors explain why two clouds of points appear at the top left and bottom right of the plot? Are these variants with exactly zero counts?

2. Additionally, it may be interesting to consider correlation in the 9-mer model by considering bins of common variants with larger counts (for instance, allele frequency >5%, >10%, 20% etc) – is Baymer expected to perform similarly well on variants under selection? (I expect that it should; they may want to comment and/or show with data)

3. Based on the power analysis presented in Table 2 (which is well appreciated!), which provides the rate of false-negatives at various effect sizes, could the authors infer the full number of true shifts in the various the 3-mer, 5-mer, 7-mer, 9-mer contexts? Can this be inferred from the number of discovered shifts (in true populations), and the false negative rate (estimated from analysis of simulated populations)?

4. Regarding the spike-and-slab prior used for regularization – its utility in preventing overfitting was nicely demonstrated. However if I understand correctly it does have at least one metaparameter (the width of the spike) and so models may benefit from optimizing this metaparameter. This is standard practice for regularized regression, usually aiming to maximize crossvalidation accuracy.

Minor:

5. cells of the last column (Population Specificity) in Table 1/supplementary Table 3 could add an asterisk in the cell indicating that this shift was or was not reported in prior work.

6. line 469: ” In all of our experiments, we focused on the entirety of the accessible, non-coding genome.: I think here "accessible" is intended to mean "accessible to sequencing", which might be confused with the meaning of "chromatin accessibility" (e.g. DHS or ATAC or such)

7. line 491: “We also compiled all sites where the high-confidence ancestral state and GRCh38 reference genome disagree, treating this collection as a call-set of derived variants.” Can they comment on whether this definition may be biased by the fact that the GRCh38 reference allele is sometimes the minor allele and not necessarily representative of the population at large?

Reviewer #2: The authors propose a new Bayesian framework for estimating the probability of observing a mutation at a site as a function of the sequence context. Their method is based on using a spike and slab prior to induce sparsity combined with a hierarchical scheme to sequentially add context to the model. They apply the method to compare polymorphism data between populations and also use it in the analysis of de novo mutations. They also conduct simulations to check that the proportional

The method is interesting and marks a significant advance in an area that has recently seen a burst of renewed interest. The paper is well-written although there are some technical details that I feel should be clarified. PLOS Genetics is an appropriate venue for this manuscript, although I think the readers of PLOS Genetics would appreciate some additional biological interpretation (the manuscript is very focused on the methodological advance).

I have two top-level concerns about the manuscript:

(1) Although the hierarchical scheme for adding context-dependence to the model makes a lot of sense from a statistical perspective, it seems quite odd from a more biological or mechanistic perspective. The model works by alternating between expanding the context at the 3' end and at the 5' end, and only allows extension to include more context if the last extension was informative. But this is quite odd physically because it means that whether context at the +3 position relative to the mutation can be included depends on whether the context at the -3 position was relevant. It would make much more sense for the inclusion of the +3 position to depend on whether the adjacent +2 position was relevant. E.g. this would make sense under the hypothesis that the effects on mutation rate are mediated by stretches of adjacent bases but that these stretches need not necessarily be exactly centered on the mutated position.

I understand that the authors have already done a great deal of work under their current scheme. But at least a spot test of e.g. expanding via a more spatially sensible hierarchy or via a symmetric widening scheme would help alleviate worries that the current scheme is leaving out important context.

(2) I think the broad readership of PLOS Genetics would be interested to hear more about the biological characteristics of the fit models and what they tell us qualitatively about the nature and extent of context dependence in human mutational dynamics. For example, if you look at a random polymorphic site, what is the distribution of the size of the context that was relevant? And what is the answer for a random (not necessarily polymorphic) position in the genome? Similarly, there is little interpretation of the actual values of phi. But there are many interesting questions, such as whether the long contexts learned by the model are indicative of sequence contexts that have a mutagenic effect (hotspots) or a protective effect. The manuscript would be improved

Minor comments:

General, but for instance Figure 2D 2E 2F y axis labels. There is a proliferation of different terminology around the parameters of the model that I found confusing. The phi are sometimes referred to as "shifts", but then also can be referred to as "edges" (as in the Figure 2 caption) or sometimes it seems are called contexts (e.g. line 214 "We note that over 61% of all sequence contexts with PIP>.95 [...]"). And then inclusion means I=1? Or can it include I=0 but phi \\neq 0?. And then differences in polymorphism rates between populations are not shifts in polymorphism rates but rather "shift differences". I suggest simplifying the terminology here to the extent that is possible.

Line 255 I was confused about the exact criteria for inclusion in Table 1.

Line 527 Is y defined?

Line 553 Does "informative" mean I=1?

Line 577 The level-by-level sampling scheme was quite unclear to me. Since there are multiple samples of the parameters at each level, how do these multiple samples play into the sampling at subsequent levels?

Reviewer #3: The manuscript introduces a new method called Baymer that can estimate mutation rates based on sequence context. It builds a hierarchical model of the sequence context and solves the model selection problem of choosing the best context size taking the data sparsity into account by using a regularizing “spike and slab” prior. The authors first use the model to investigate differences in the sequence context of polymorphisms between populations. Secondly, they explore using polymorphisms to predict the rate of de novo mutations.

A few other recent articles have looked at methods for building 9mer models of mutations, but this is a new approach to doing so. I like the idea of the grafted tree "transfer learning" model where de novo mutations are used to train the upper levels in the hierarchical model, but polymorphism data is used to train the differences between the longer k-mers. Overall the manuscript is well-written and interesting.

Specific comments:

You write that a limitation of the model is the treatment of multi-allelic sites because you assume that you only see one mutation at each site. I think that not considering recurrent mutations is a bigger problem than multi-allelic variants when you train on the polymorphism data. In a large population sample like gnomAD the majority of all possible CpG mutations are polymorphic, and because of the high mutation rate, there will have been recurrent mutations a many CpG sites. In line 340, you write that you were surprised that models trained on singletons or doubletons did not do better. I do not think this is so surprising - if you do not take the low number of CpG sites that are not already polymorphic into account you would significantly underestimate their mutation rate using the singletons. Was the model fit with singletons especially bad for CpG transitions? Either increasing the mutation counts to account for recurrent mutations or reducing the background context count to reflect that you cannot observe a singleton at an already polymorphic site should improve the models.

I tried installing the software using the included install script, but it failed because conda could not find many of the specific package versions specified in the yaml file. Maybe you need to specify which conda channels should be used?

I was able to get it installed by using other versions of the packages than the ones specified in the conda yaml file, and I tried running the code in “baymer_tutorial.ipynb”. But It looks like the tutorial in the notebook is not up to date with the names of the parameters in the functions it calls. When I tried counting mutations by running the second cell in the notebook, it gave me this error:

“TypeError: driver() missing 1 required positional argument: ‘quality_bool’”

In the 4th cell I got this:

TypeError: driver() got an unexpected keyword argument ‘random_seed’

In the 5th cell:

TypeError: driver() got an unexpected keyword argument ‘param_config_file’

Can you say something about what the running time of the method is and whether it could be used on 11mers?

Reviewer #4: Summary:

The authors present a new model and computational implementation called Baymer for inferring polymorphism probability (a proxy for mutation rate) from nucleotide sequence context. Whereas most previous work has focused on fixed local sequence context (usually 3-mers), the key insight of Baymer is that related mutation contexts (e.g. two 5-mers with a common central 3-mer) may be expected to have similar polymorphism probabilities. They construct a hierarchical tree of increasing context size around a root singlet mutation type, and model the log probability of polymorphism as additive over parameters associated to tree edges. This parameter sharing of related contexts allows more reliable estimation, especially for larger contexts that are sampled very sparsely. They also introduce explicit sparsity assumptions on the edge weights via a spike-slab prior. This parameterization and regularization results in a compressed representation of polymorphism probability over a range of context sizes, and is more robust than empirical counts in terms of several out-of-sample prediction tasks.

Overall, I found the methods and results to be compelling, creative, rigorously validated, and mostly described well. Some aspects are somewhat heuristic and less interpretable, but overall I think the work shows that the model is behaving sensibly and improving on simpler baselines, although there are some directions for improvement.

Major comments:

As the authors note, the alternating addition of 5’/3’ nucleotides across tree levels is somewhat awkward and not directly interpretable. It should also be noted that this excludes half of all even-ordered contexts. For example, as shown in figure 1, level 2 captures dinucleotide mutation types like A*>C* by adding nodes for the 4 possible 3’ contexts of the focal site. However, IIUC there are no nodes in any tree level that correspond to dinucleotide mutation with 5’ context, e.g. *A>*C. This asymmetry seems to omit a large number of mutation types from consideration, and also seems inconsistent with the approach of collapsing variants by reverse complement (with all focal site ancestral states collapsed to A or C). Are results robust to swapping the alternation order?

As a side remark related to the above point, I wonder if it’s possible to augment the tree with nodes for both 5’ and 3’ additions, and sum over paths (the tree would become a DAG). For example, TCC>T would be the child of both TC>T and CC>T, and the probability contributions could be additive over such parallel paths.

One notable limitation in the Baymer approach is that it doesn’t allow parameter sharing among mutation types that have the same ancestral state and context, but different derived states. Indeed, a separate parameter tree is constructed for each singlet mutation type. This may limit the ability to detect signatures of mutability for highly specific ancestral contexts with less specified derived states (i.e. parameter sharing for mutation targeting to identical/similar contexts, but different derived states).

Polymorphism probability is being interpreted as a proxy for mutation rate. To test this, it would be very interesting to perform a simulation study using a population genetic simulator (like msprime or slim), where the mutation rate can be parameterized according to the tree-based parameters and a standard demographic model. Even just asking how well the polymorphism-based estimates reflect the underlying mutation trees would be interesting, let alone additional modeling work to put the trees into a history inference procedure. As the authors note in the discussion, this is left for future work, so not a necessary addition for this work. But it may be straightforward to assess at some point how good a proxy for mutation rate is the polymorphism probability.

Minor comments:

Instead of the even-odd basepair test-train split, I would have thought windowed region (or whole chromosome) holdouts would be a natural way to hold out SNPs for testing. Did the authors consider this?

I didn’t really follow the arguments about a possible 2nd signal in the TCC>T signature with a distinct CC>T subcomponent. Given the asymmetric leveling, isn’t it impossible in this model to have found the 5’ dinucleotide subcomponent TC>T?

It is not clear if SNPs and mutation context are polarized with respect to an inferred human ancestral state, or taken simply as REF/ALT states.

The mutation types *AC>*CC was previously shown to be enriched in the 1KG Japanese population (Harris and Pritchard), but this was later found to be a likely cell line artifact (Anderson-Trocmé et al.). I did not see this artifactual signature in Baymer results—can you comment on why?

It is not clear how equation 1 is constrained to the unit interval (i.e. how it represents a probability). Given the priors on the phi parameters, apparently there is a nonzero probability that p will exceed unity. Similarly, elements of the p vector in equation 2 could exceed unity. How is this being handled?

It’s great that the authors have made their code repository available on GitHub. Unfortunately, the install procedure failed on both linux and mac for me. I was able to install after editing the conda environment file heavily (removing many packages, removing all the version specifications, and removing the user-specific prefix). The tutorial notebook then failed due to a API usage error in `baymer.mutation_counter.driver()`. I suggest the authors revise the install procedure to be replicable across machines/users with minimal dependencies, and debug the tutorial notebook.

**Have all data underlying the figures and results presented in the manuscript been provided?**

Reviewer #1: Yes

Reviewer #2: None

Reviewer #3: None

Reviewer #4: Yes

PLOS authors have the option to publish the peer review history of their article (what does this mean?). If published, this will include your full peer review and any attached files.

Reviewer #1: No

Reviewer #2: No

Reviewer #3: **Yes: **Søren Besenbacher

Reviewer #4: **Yes: **William DeWitt

---

## [Decision Letter · Decision Letter 1]

1 Jun 2023

Dear Dr Voight,

We are pleased to inform you that your manuscript entitled "Regularized sequence-context mutational trees capture variation in mutation rates across the human genome" has been editorially accepted for publication in PLOS Genetics. Congratulations!

However, two reviewers have raised concerns with the software. We do not wish to delay the process any further with a minor revision decision, it's in your interests to try to resolve these problems before the paper is published which we urge you to do and make any associated adjustment to the text when preparing the final version for publication.

Yours sincerely,

David Balding

Section Editor

PLOS Genetics

Reviewer's **Comments to the Authors:**

Reviewer #1: The authors have answered my queries satisfactorily. The study was improved by the revisions.

Reviewer #2: The authors have provided a thoughtful response to reviewer comments and made many improvements to the manuscript. These include a new main text figure (Figure 2) exploring different schemes for expanding local sequence content and several new passages focused on biology interpretation, particularly the new passage on comparison of mutation rates among the great apes. Overall these changes have substantially improved the manuscript and sufficiently addressed my concerns from the previous round of review.

Reviewer #3: The authors have adressed my comments to the text but I still had problems installing the tool and running it from the command line.

When I try to install the conda environment from the yaml file I get an error saying that the environment requirements cannot be solved:

Could not solve for environment specs

Encountered problems while solving:

- package numba-0.53.1-py39ha9a9b91_0 requires numpy >=1.19.5,<1.21.0a0, but none of the providers can be installed

The environment can't be solved, aborting the operation

I then installed the required packages without specifying specific version numbers and was able to run the first two commands in the tutorial but the “generate_count_json.py” command fails with the following error:

File "/opt/miniconda3/envs/baymer/lib/python3.9/site-packages/numpy/core/_methods.py", line 49, in _sum

return umr_sum(a, axis, dtype, out, keepdims, initial, where)

TypeError: can only concatenate str (not "int") to str

Reviewer #4: In this revision, the authors have added new analyses that strengthen and validate the model and results, and they have made modifications to the text to improve clarity and correctness. My main concerns and comments from the previous version have been adequately addressed. The authors directly address the more heuristic aspects of the model, showing results hold up to varying these.

There are still some issues with usability of the software (installation and tutorial notebook), and it would be nice if those could be rectified. I'll summarize some difficulties:

- The install.sh bash script is error-prone, because if the Conda env creation fails for any reason, it will still attempt to install Baymer on the last line, and this would then install it in the base environment, instead of the Baymer environment as intended. This could be avoided by using the && operator in bash, so that subsequent commands are not run after one fails.

- The current "python setup.py install" line raises a deprecation warning: "SetuptoolsDeprecationWarning: setup.py install is deprecated. Use build and pip and other standards-based tools". It is probably preferable to install via pip with "pip install .", where "." refers to the root directory. This could even be bundled into the Conda env file, so that no wrapper shell script is needed.

- Install failed for me because Conda could not resolve the version numba=0.53.1. You might want to remove or weaken the version specifiers (I was able to install successfully after removing them all).

- I’ve attached a modified Conda environment yaml file that I used to install Baymer and its dependencies, with changes addressing the issues above. I also added a jupyter dependency, so I could try running the tutorial notebook.

- Cell 3 of the tutorial notebook issues two warnings, one about a deprecation, and the other about a possibly invalid DataFrame operation. Maybe these can be modified, or at least checked for possible bugs.

- The notebook then fails to run due to FileNotFoundError errors beginning on cell 5. I suspect the notebook is relying on the contents of a local working directory that has diverged from what’s currently on GitHub. I suggest checking all this on a fresh clone of your repo. Better yet, you could set up a GitHub action to automatically check that install and notebook execution works, and have it run any time you push to the main branch.

**Have all data underlying the figures and results presented in the manuscript been provided?**

Reviewer #1: Yes

Reviewer #2: Yes

Reviewer #3: None

Reviewer #4: None

PLOS authors have the option to publish the peer review history of their article (what does this mean?). If published, this will include your full peer review and any attached files.

Reviewer #1: No

Reviewer #2: No

Reviewer #3: No

Reviewer #4: **Yes: **William DeWitt

**Data Deposition**

http://datadryad.org/submit?journalID=pgenetics&manu=PGENETICS-D-22-01424R1

**Press Queries**

---

## [Editor Report · Acceptance letter]

26 Jun 2023

PGENETICS-D-22-01424R1 

Regularized sequence-context mutational trees capture variation in mutation rates across the human genome 

Dear Dr Voight, 

We are pleased to inform you that your manuscript entitled "Regularized sequence-context mutational trees capture variation in mutation rates across the human genome" has been formally accepted for publication in PLOS Genetics! Your manuscript is now with our production department and you will be notified of the publication date in due course.

With kind regards,

Jazmin Toth

PLOS Genetics

On behalf of:
